# Design and On-Orbit Performance of Ku-Band Phased-Array Synthetic-Aperture Radar Payload System

**DOI:** 10.3390/s24206741

**Published:** 2024-10-20

**Authors:** Wei Yan, Xiaomin Tan, Jiang Wu, Mingze Yuan, Hongxing Dang, Wujun Chang

**Affiliations:** 1Xi’an Institute of Space Radio Technology, Xi’an 710100, Chinawuj74@cast504.com (J.W.); yuanmingze@cast504.com (M.Y.);; 2Beijing Minospace Technology Co., Ltd., Beijing 100089, China; chang3517@163.com

**Keywords:** Ku band, SAR system, payload, Taijing-4(03), on-orbit, application, micro-nano, satellite

## Abstract

The current emphasis in the advancement of space-based synthetic-aperture radar (SAR) is on lightweight payloads under 100 kg with resolutions surpassing 1 m. This focus is directed toward meeting the launch criteria for multiple satellites on a single rocket and cutting costs. This article discusses the creation and progress of a Ku-band SAR payload for the Taijing-4(03) satellite, launched on 23 January 2024 and accompanied by four other satellites. The SAR payload design was customized to meet the demands of a micro-nano satellite platform, resulting in a lightweight, flat design weighing less than 80 kg, seamlessly integrated with the plate-shaped satellite platform. The article also introduces a beam optimization strategy for the phased array SAR antenna, significantly boosting the SAR system’s performance. The SAR payload provides various operating modes like slide-spot, strip, Scan 1, Scan 2, and others, with a maximum achievable resolution exceeding 1 m. Extensive in-orbit testing of the payload produced numerous high-quality SAR images with potential uses in emergency disaster mitigation, safeguarding ecosystems, monitoring forests, managing crops, tracking sea ice, and more.

## 1. Introduction

In the past few years, there has been a notable rise in the advancement of commercial aerospace, resulting in an increasing emphasis on micro-nano synthetic-aperture radar (SAR) satellite constellations among prominent aerospace powers. Currently, numerous large-scale micro-nano SAR satellite constellations have been deployed worldwide [1,2,3,4]. These constellations, characterized by high temporal and spatial resolution capabilities, are poised to revolutionize the future of microwave remote sensing applications. Micro-nano satellite satellite constellations offer several advantages, including low cost, rapid development cycles, quick response times, near real-time observation capabilities, and promising potential for civil and commercial applications [5,6,7].

There are two common micro-nano SAR satellite technology systems: a phased array antenna system and reflector antenna system [8,9,10]. Spaceborne SAR with a reflector antenna system offers the advantages of a large antenna area, high resolution, lightweight, and low cost. However, it also presents challenges such as inflexible beam scanning, as seen in systems like Capella constellation [11,12,13,14] and Umbra constellation [15,16] in the United States. Phased array spaceborne SAR, on the other hand, boasts flexible beam scanning and high antenna efficiency, albeit at a relatively higher cost. Examples of typical systems include the ICEYE constellation in Finland [17,18,19].

The current large phased array SAR systems exhibit a high performance, boasting resolutions better than 0.5 m and offering a variety of operational modes. However, their weight typically exceeds 1 ton, necessitating the use of large launch vehicles such as the German TerraSAR-X satellite [20,21] and the Italian CSG satellite constellation [22,23,24]. In the past few years, there has been significant progress in the advancement of micro-nano phased array SAR satellites, which have a weight of under 100 kg. Despite this progress, these satellites face limitations in terms of their range of incidence angles, which usually do not exceed 35°, and their system sensitivity is only −15 dB. This falls short of meeting the application requirements, with the ICEYE constellation being a typical example.

Taijing-4(03) is a high-resolution flat-panel SAR imaging satellite developed based on the MN200S platform by the Beijing Minospace technology company. It is also the world’s first Ku-band phased array radar imaging satellite. The SAR payload of the satellite features a highly integrated design, weighing less than 80 kg, with a peak transmission power exceeding 4000 W, achieving a resolution better than 1 m and a system sensitivity of −18 dB. The satellite was successfully launched on 23 January 2024 as part of a mission involving five satellites. Following deployment in orbit, the satellite captured a series of high-quality SAR images that have been instrumental in diverse applications such as emergency disaster relief, ecological conservation, sea ice monitoring, and target recognition.

Section 2 contains the work modes and system design of the Ku-band SAR for Taijing-4(03). Section 3 describes the antenna model of the SAR payload, followed by the SAR on-orbit performance and application in Section 4. The conclusion is given in Section 5.

## 2. Sar Payload Design

### 2.1. Overview

A Ku-band SAR payload was designed based on the resolution and swath requirements. The key parameters of the SAR payload are outlined in Table 1.

Pulse width is mainly related to the average transmission power, peak transmission power, and satellite energy. When the satellite energy is limited, the peak transmission power of Ku-band SAR is 4000 W. When the pulse repetition frequency of the transmission is fixed and when the transmission pulse width is too narrow, the average transmission power will be low, resulting in an NESZ lower than −18 dB, thus affecting the image quality.

The bandwidth of the system is related to the distance resolution. A total of nine bandwidths are designed for the system, of which 600 MHz, 540 MHz, and 500 MHz are used for spot mode and sliding spot mode to achieve a resolution better than 1 m in the ground distance direction. Frequencies of 240 MHz, 200 MHz, 180 MHz, and 150 MHz are used for strip mode to achieve a resolution better than 2m in the ground distance direction. The frequencies 40 MHz and 20 MHz are used for Scan 1 mode and Scan 2 mode, respectively, to achieve a resolution better than 10 m and 20 m in the ground distance direction.

The number of phased array panels is related to the aperture size of the phased array antenna. In order to ensure that the system’s NESZ is better than −18 dB, according to the radar equation, the aperture of the phased array antenna is 3.6 m × 0.6 m. The antenna and the satellite platform are designed to be integrated, and the size of the satellite platform is 2 m × 0.7 m. The Ku-band SAR antenna is divided into five panels; each panel is 0.72 m × 0.6 m in size. The central three panels are installed on the satellite body. At the same time, considering the envelope constraint of the rocket, the antenna needs to be folded and unfolded after being in orbit. Therefore, the antenna is designed to be composed of five sub-boards, and the three in the middle are fixed to the satellite platform, as shown in the figure. The sub-boards on the left and right sides are in orbit or unfolded, as shown in the figure.

The energy of the satellite is related to the power consumption of the SAR system. In this system, the peak transmission power of the SAR system is 4000 W and the maximum duty cycle is 20%, so the corresponding power consumption is 3200 W, which puts higher requirements on the energy of the satellite platform. Taking into account the performance indicators of the SAR payload and the weight limitation of the satellite (<230 kg), the satellite chooses a system for power supply, which can meet the SAR payload power consumption requirement of 3200 W.

The satellite is in a stowed configuration before launch, as shown in Figure 1. The panels on both sides of the antenna are deployed after being in orbit, and the satellite configuration after deployment is shown in Figure 2. Considering that the antenna works in a 10:30 a.m. orbit at the descending node time, the external heat flow of the space environment changes violently, so the antenna adopts intelligent temperature control measures to ensure the stability of the antenna’s on-orbit temperature.

The Taijing-4(03) satellite was launched on 23 January 2024 as part of a single rocket carrying five satellites aboard the Lijian-1 Yao-3 rocket. The satellite utilizes a side-hung configuration for docking with the rocket, depicted in Figure 3. The Lijian-1 rocket, with dimensions of 30 m in length, 2.65 m in diameter, and a liftoff weight of 135 metric tons, showcases impressive engineering capabilities. Particularly noteworthy is its proficiency in payload delivery, capable of transporting multiple SAR satellites totaling up to 1.5 tons to a sun-synchronous orbit situated approximately 500 km above Earth.

### 2.2. Working Modes

There are five main operating modes for the Ku-band SAR payload, including spot mode, sliding spot mode, strip mode, Scan 1 mode, and Scan 2 mode, as shown in Figure 4. The sliding spot or spot mode is used for the high-resolution monitoring of artificial targets, which is discontinuous in the azimuth direction. The sliding spot is mainly used for the high-resolution monitoring of single targets such as airports and ports. The obtained resolution is better than 1 m, and the scene width reaches 20 ×10 km, which can completely cover the mission area. Compared with the sliding spot, the spot mode has a scene width of only 5 × 5 km, but the data volume is 1/8 of the sliding spot, which is easier to transmit to the ground through the satellite data transmission channel. The imaging time is only 6 s, which can be used to cover multiple point targets. The strip mode is used for high-resolution continuous observations of urban rivers, surface coverage, etc., and can be used for emergency rescue and other applications. Scan 1 mode and Scan 2 mode are mainly used for low-resolution applications, including forest change monitoring, glacier monitoring, typhoon monitoring, and other applications.

Spot mode and sliding spot mode can achieve a high azimuth spatial resolution through antenna scanning in azimuth, but it will cause discontinuity in the azimuth width; for example, the azimuth width of spotlight mode is only 5 km. This mode is realized by transmitting a wideband signal in the range direction, and the maximum bandwidth of the range direction signal is 600 MHz. In the striped mode, the system can achieve the target of 2 m resolution and 20 km width, and the maximum range signal bandwidth is 240 MHz. In Scan 2 modes, the system can achieve the target of 20 m resolution and 200 km width, and the maximum range signal bandwidth is 40 MHz.

### 2.3. System Design

Ku-band SAR is composed of three parts: the signal processor, transceiver channel, and phased array antenna, as shown in Figure 5. Considering the lightweight requirements of the satellite platform for the payload, the SAR payload is designed with high integration. The waveguide slot array of the SAR phased array antenna adopts a thin-walled structure design, and combined with a carbon fiber structure frame, the antenna surface density is greatly reduced. The transceiver channel has the functions of a transmitting channel, receiving channel, and internal scaler at the same time and is packaged in a single machine, which improves the integration level. The signal processor adopts a system-on-a-chip (SOC) structure and integrates chips such as ADC and DAC, which effectively improves the ability of signal generation, acquisition, and processing.

The SAR payload utilizes a retractable flat active phased array antenna measuring 3.6 m in width by 0.6 m in height. The antenna comprises 600 Transmit–Receive (TR) channels, arranged into 15 vertical columns and 40 horizontal rows in a configuration spanning across five panels. The antenna has a two-dimensional beam scanning capability, the scanning angle range in the azimuth direction is ±1.5°, and the scanning angle range in the pitch direction is ±15°, which can meet the needs of antenna beam scanning in different SAR operating modes. The antenna contains a total of 150 TR modules; each TR module contains four channels, and the peak transmit power of each channel is greater than 6.8 W, so the total system transmit peak power is greater than 4000 W.

After launching and orbit adjusting, a deployable mechanism deploys the panels from a stowed launch configuration into its deployed in-orbit configuration. Before SAR operation, the antenna has to point to −32° (right-looking) or +32° (left-looking), relying on the roll steering of the satellite platform. This dual-side looking capacity is valuable in reducing the revisit interval time. Zero Doppler steering, which employs yaw and pitch steering to eliminate the Doppler center frequency shift, was introduced to the satellite.

## 3. Antenna Model

### 3.1. Array Synthetic Pattern Model

The antenna can be segmented into multiple subarrays, with each subarray linked to a single TR element. The antenna consists of M×N TR elements, where *M* signifies the quantity of elements in the elevation dimension and *N* indicates the azimuth dimension.

The spatial distribution of the radiation pattern from a satellite-based phased array antenna is determined by
(1)$$\begin{array}{cc} F\left(\theta_{El},\theta_{Az}\right) & =\sum_{m=0}^{M-1}\sum_{n=0}^{N-1}(P_{SA}\left(\theta_{El},\theta_{Az}\right)\times A_{mn}\times E_{mn}\\ \; & \times \exp^{j\frac{2\pi }{\lambda }\sin\theta_{zl}\cos\theta_{Az}\left(n-1\right)\Delta y}\times \exp^{j\frac{2\pi }{\lambda }\cos\theta_{zI}\sin\theta_{AL}\left(m-1\right)\Delta x}) \end{array}$$
where by the structure involves the angle of elevation $\theta_{El}$ and azimuth $\theta_{Az}$, as well as the subarray configuration $P_{SA}\left(\theta_{El},\theta_{Az}\right)$. The numerical order of the subarrays in both the vertical and horizontal directions is indicated by sequence numbers *m* and *n*. The spacing between subarrays in terms of the height Δx and horizontal distance Δy is also measured. The operative wavelength is denoted as λ. The $\theta_{El}$ and $\theta_{Az}$ represent the SAR antenna’s boresight elevation beam direction and azimuth beam direction, respectively. For the spaceborne SAR system, the antenna scanning at $\theta_{El}$ corresponds to the beam switching in the range direction, which is mainly used in strip and scanning modes, and scanning at $\theta_{Az}$ corresponds to the beam switching in the azimuth direction, which is mainly used in the sliding beam mode.

The configuration consists of the angle of elevation $\theta_{El}$ as well as the azimuth angle $\theta_{Az}$ and the subarray design $P_{SA}\left(\theta_{El},\theta_{Az}\right)$. The numerical order of the subarrays in both the vertical and horizontal directions is given by the sequence numbers *m* and *n*. The distances between the subarrays in the vertical and horizontal directions are denoted as Δx and Δy, respectively. The operational wavelength is λ.

The stimulation factors of the phased array antenna are computed as [25]
(2)$$A_{mn}=a_{mn}\times \exp^{j\varphi_{mn}}$$
where the weight of the amplitude is $a_{mn}$ and phase is $\varphi_{mn}$.

The coefficients for the error calculation $E_{mn}$ can be derived from the computed antenna network, as depicted in Figure 6. The one, two, and three mentioned in the figure represent the loop of the internal calibration signal, which indicates the signal path of the transmission calibration, reception calibration, and reference calibration.

The transmission calibration path is 1-2-3-6-7-8-9-3-4-5, which can obtain the amplitude and phase of the transmission channel of each TR component of the antenna, where the number 1 indicates that the signal enters the Tx/Rx Channel after coming out of the DA. Through transmit calibration, the system can obtain the amplitude and phase of each TR component to correct the antenna transmit pattern.The path of receive calibration is 1-2-3-9-8-7-6-3-4-5, which can obtain the signal amplitude and phase of each TR component receiving channel of the antenna. Through transmit calibration, the system can obtain the amplitude and phase of each TR component to correct the antenna receive pattern.The path of reference calibration is 1-2-3-4-5, which can obtain the amplitude and phase of the system except the antenna, which is used to correct the signals of transmit calibration and receive calibration.

The core of optimizing antenna patterns involves fine-tuning the excitation coefficients for the TR component of the antenna. When broadening the antenna beam, the antenna’s gain diminishes, resulting in the distortion of the antenna pattern and consequently impacting the SAR system’s performance. Hence, optimizing the antenna pattern alongside the SAR system is crucial for determining the best weight value.

The precise electrical model of the antenna offers benefits not only for beam optimization but also for reducing the testing time of the antenna patterns, considering the large number of beams resulting from the five observing modes. It would be advantageous to conduct some experiment modes in orbit in the future. The performance of experiment modes can be evaluated using this antenna model. Given the strong consistency, antenna pattern simulation and configuration code generation can be performed on the ground as the antenna patterns cannot be practically tested post-satellite launch.

### 3.2. Beam Optimization Technique

The technique for beampattern optimization is illustrated in Figure 7. The initial population is generated based on the Antenna Parameter, mainly including the initial amplitude $a_{mn}$ and phase $\varphi_{mn}$ of each TR component of the antenna. Fitness is the fitness function for generating the antenna pattern calculation based on the amplitude and phase of the antenna TR component. If Fitness meets the requirements, the optimization stops; otherwise, it enters the Selection Crossover stage. This stage cross-selects the amplitude and phase of the TR component with a certain probability. After the cross-selection is completed, it enters the Fitness evaluation. If it still does not meet the requirements, it enters the Mutation stage. This stage represents the amplitude and phase of the TR component in binary and performs mutation operations on some values.

The differences between this method and the conventional antenna beam optimization method based on a genetic algorithm are mainly in two aspects: one is the correction of the fitness function and the other is the change in the selection of the initial weight of the antenna. Instead of relying on random amplitude and phase weights, the initial weight of the antenna is determined by a set of weights derived from the anticipated pattern of the antenna, facilitating a quicker convergence rate.

The evaluation criteria for the antenna pattern can be categorized into the primary lobe region and secondary lobe region. In the primary lobe area, the evaluation criteria consist of the system sensitivity (NESZ) [22]. As for the secondary lobe area, the evaluation criteria include system azimuth ambiguity (AASR) and range ambiguity (RASR) [23].

All the aforementioned objective evaluation criteria utilize a non-linear exponential format, meaning that a lower calculated value of the criteria indicates a more favorable outcome. The evaluation criteria comprise three components, which can be represented as
(3)$$fit=w\times \left|patt_{t}-patt_{d}\right|$$

In this system, the optimized antenna pattern is denoted as $patt_{t}$, while the anticipated antenna pattern is referred to as $patt_{d}$. The antenna pattern is measured in decibels (dB), and the system’s weight is denoted as *w*.

The system’s fitness function can be represented as
(4)$$fit=fit_{NESZ}+fit_{flat}+fit_{AASR}+fit_{RASR}$$

The fitness functions are influenced by the NESZ and flatness, while the fitness functions are impacted by the system azimuth ambiguity $fit_{AASR}$ and range ambiguity $fit_{RASR}$.

Given that the anticipated trend follows the pattern $patt_{d}\left(\theta_{El},\theta_{Az}\right)$, spatial sampling for various antenna azimuth and elevation angles can be conducted using the following equation:(5)$$\theta_{El}\left(k\right)=\left\{\begin{array}{c} \arcsin\left(\frac{k\lambda }{M\times \Delta y}\right),k\in \left[0,M/2\right]\\ \arcsin\left(\frac{k\lambda -2M\times \Delta y}{M\times \Delta y}\right),k\in \left(M/2,M\right] \end{array}\right.$$
(6)$$\theta_{Az}\left(l\right)=\left\{\begin{array}{c} \arcsin\left(\frac{l\lambda }{N\times \Delta x}\right),l\in \left[0,N/2\right]\\ \arcsin\left(\frac{l\lambda -2N\times \Delta x}{N\times \Delta x}\right),l\in \left(N/2,N\right] \end{array}\right.$$

The original weight of every individual TR component of the antenna is $A_{mn}$. The expected pattern can be reversed by carrying out the necessary actions at the designated sampling locations:(7)$$A_{mn}=\sum_{k=0}^{M-1}\sum_{l=0}^{N-1}patt_{d}\left(\theta_{El}\left(k\right),\theta_{Az}\left(l\right)\right)\exp^{j\frac{2\pi }{M}\times k\times m}\exp^{j\frac{2\pi }{N}\times l\times n}$$

The initial weight of each TR can be determined using Formula (Equation 7).

In space-based SAR systems, maintaining uniformity in the coverage area requires widening the beam when facing a narrow incident angle, with a magnification factor of two times the initial width. The design of the antenna’s beam optimization was implemented utilizing the methodology detailed in this paper, and the antenna’s radiation pattern was simulated and validated, as depicted in Figure 8.

Based on the comparison between simulated and actual data, the alignment of antenna patterns is fairly strong, with the main lobe experiencing an error of less than 0.5 dB, thus meeting the specifications of the SAR system.

### 3.3. Beam Optimization Result

Compared with the spaceborne phased array SAR mentioned in other references (ICEYE [17,18,19] and TerraSAR [26,27,28]), this paper mainly combines the beam optimization technology of system design. The comparison of SAR key parameters is listed in Table 2. Its main feature is that it can achieve a wider mapping swath at the same resolution. The beam optimization design method used in this paper mainly improves the mapping swath of the near-end incident angle and can increase the mapping swath of each burst from 15 km to 25 km. In Scan 2 mode, the system uses 10 bursts for imaging, of which six sub-bursts are involved in the optimization, so the system width is increased from the traditional 20 m/140 km to 20 m/200 km.

After adopting the above optimization method, the spaceborne SAR system was greatly improved, especially for Scan 2 mode, as shown in the figure below. Scan 2 mode is implemented by 10 burst scans. With the traditional beam design method, the system has a remote ground distance of 380 km, while with the improved optimization method proposed in this paper, the system has a remote ground distance of 440 km, so the image width was increased by 60 km, as shown in Figure 9.

## 4. Sar On-Orbit Performance

### 4.1. SAR Performance

The NESZ is related to the system sensitivity to low radar backscatter areas. The NESZ corresponds to the scattering coefficient for which the signal-to-noise ratio (SNR) is equal to one. After the SAR payload is in orbit, the antenna pattern and system sensitivity are evaluated. Antenna patterns are mainly evaluated using the Amazon rainforest. The SAR payload works in imaging mode and noise calibration mode. The Amazon rainforest image is corrected using the noise power data of the noise calibration mode and the normalized backscatter coefficient of the Amazon rainforest, thereby obtaining the on-track curve of the NESZ, as shown in the following formula:(8)$$\mathrm{NESZ}=\sigma_{R}-\left(P_{r}-P_{N}\right)$$

Here, $P_{r}$ is the echo power and $P_{N}$ is the noise power. The normalized backscatter coefficient of the Amazon rainforest is taken as −6.5 dB, which comes from the measurement results of the China HY-2 scatterometer.

The evaluation results of the system sensitivity are shown in Figure 10. The X-axis means the range of sample points in this scene. Figure 10a shows the power received by the satellite in imaging mode, Figure 10b shows the noise power received by the satellite in noise calibration mode, and Figure 10c is the NESZ result analyzed according to Formula (8). According to the evaluation results, the sensitivity of the system at the center of the antenna beam reaches −20.0 dB, which meets the design requirements of the system. Considering that the on-orbit test was carried out in the early stage of the on-orbit performance, the antenna beam pointing is biased and was not corrected, so there is an asymmetry in the figure, but it does not affect the evaluation of system sensitivity. In this mission, the orbital altitude of the satellite is 524 km, and the corresponding slant range is 601 km. Considering that the atmospheric attenuation effect on the Ku band is less than 0.1 dB, the atmospheric attenuation is not considered.

Antenna patterns are mainly evaluated using the Amazon rainforest. The payload evaluates the antenna pattern based on the results of satellite imaging by irradiating the Amazon rainforest with different beams. Considering that there are more than 5000 beams on board the satellite, only some antenna beams were evaluated during the on-orbit test. The result of the antenna pattern is shown in Figure 11. According to the results in the figure, the antenna pattern of the satellite in orbit is consistent with the ground test results, and the error is less than 0.5 db. Considering that all beams cannot be tested after the satellite is in orbit, it is necessary to evaluate all beams of the antenna according to the on-orbit test results of the wavelength division beam and combined with a high-precision antenna model.

### 4.2. On-Orbit Image

The Taijing-4(03) satellite was launched on 23 January 2024. The first synthetic-aperture radar (SAR) observation took place on 27 January 2024, as shown in Figure 12. Following this, in-orbit performance evaluation and product application tests were conducted incrementally. Taijing-4(03) has been officially operational since April 2024.

The system is capable of achieving azimuth continuous imaging with a 2 m resolution in strip mode, with a range width of up to 20 km. This allows for medium- and high-resolution imaging of global land. The imaging results from strip mode show well-defined features such as runways and covered bridges at airports, urban buildings, water bodies, mountains, crops, and forests, with clear structures. These results suggest that the model holds great potential for the aforementioned applications.

The SAR system can achieve better than 1 m resolution imaging in sliding spotlight mode, the range width is 10 km, and the azimuth width is 20 km. It can be used for the high-resolution imaging of local areas. The Francis Scott Key Bridge was a steel arch-shaped continuous truss bridge, the second longest in the United States and third longest in the world. On 26 March 2024, at 05:28 UTC, the main spans of the bridge collapsed due to a collision caused by the loss of power of the Singapore-registered container ship MV Dali, which struck the southwest supporting pier of the main truss section. The SAR image is photographed at 16:19 on 30 March 2024, as shown in Figure 13. According to SAR images, the bridge crashed into four sections, and a large number of ships were rescuing and repairing around it.

The SAR payload can be designed with Scan 1 mode and Scan 2 mode in order to meet the user’s needs for low resolution and a wide mapping band, among which Scan 1 mode can achieve a 10 m resolution and 70 km width. Scan 2 mode can achieve a 20 m resolution and 200 km width. The payload of the Amazon rainforest was imaged using Scan 1 mode, as shown in Figure 14. This imaging mode can be used for the normalized monitoring of the Amazon rainforest for monitoring applications such as deforestation and forest fires.

The depletion of the Antarctic ice sheet, a significant source of Earth’s freshwater, contributes directly to the increase in global sea levels and the freshening of the Southern Ocean. Research using observations and simulations has shown that the melting of ice shelves at their base, caused by the penetration of warm water onto the continental shelf of Antarctica, has a crucial impact on the overall mass equilibrium of the ice sheet. The Antarctic glacial tongue was observed by the SAR system in March, 2024. With a width of 350 × 200 km, the scene can completely cover the Antarctic ice tongue. From the results in Figure 15, we can see the shape of the ice tongue, which can be used to further study the melting state of Antarctic ice and snow.

## 5. Conclusions

In this study, a Ku-band spaceborne synthetic-aperture radar payload is proposed for the first time in the world. The payload weighs less than 80 kg and features a flat low-profile design integrated with the Taijing-4 03 satellite. This design allows for the potential establishment of a future constellation where multiple satellites can be launched simultaneously. The payload is lightweight and has a low profile, a high resolution, and various operation modes, making it highly promising for applications. The Taijing-4 03 satellite was successfully launched on 23 January 2024, equipped with the Ku-band SAR payload. In orbit, the SAR system has captured a large number of high-quality images with a resolution exceeding 1 m and a coverage width of up to 200 km. Its capabilities extend to high-resolution emergency observations of hot spots as well as wide-area information acquisition in areas such as forests, oceans, and polar regions.

## Figures and Tables

**Figure 1 sensors-24-06741-f001:**
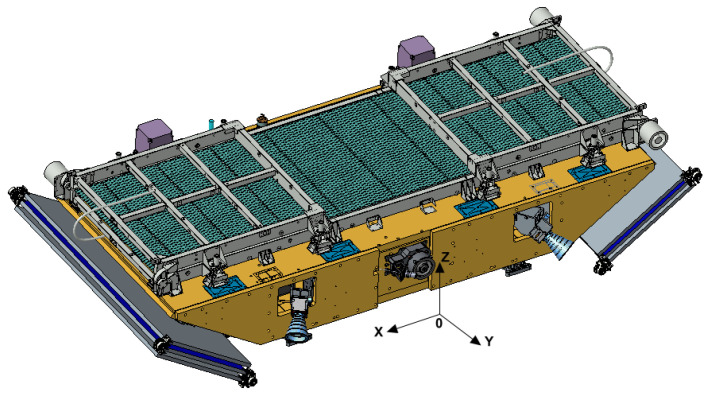
Satellite stowed configuration.

**Figure 2 sensors-24-06741-f002:**
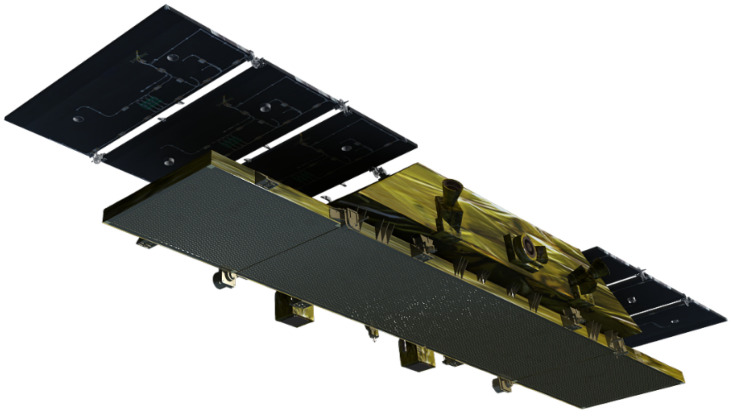
Satellite deployed configuration.

**Figure 3 sensors-24-06741-f003:**
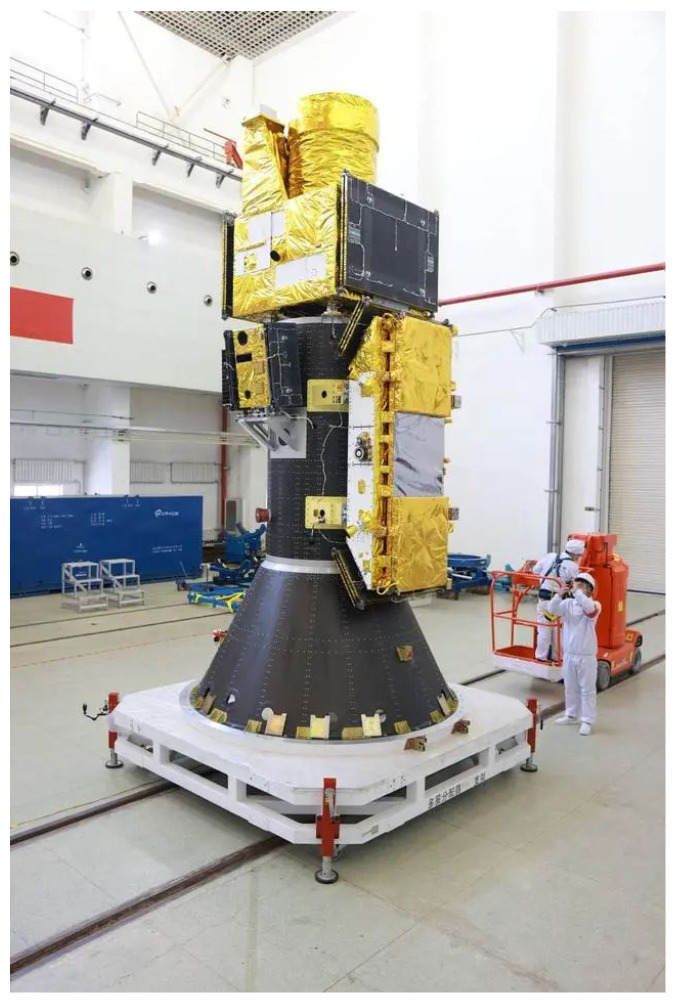
Satellite side-hung docking configuration.

**Figure 4 sensors-24-06741-f004:**
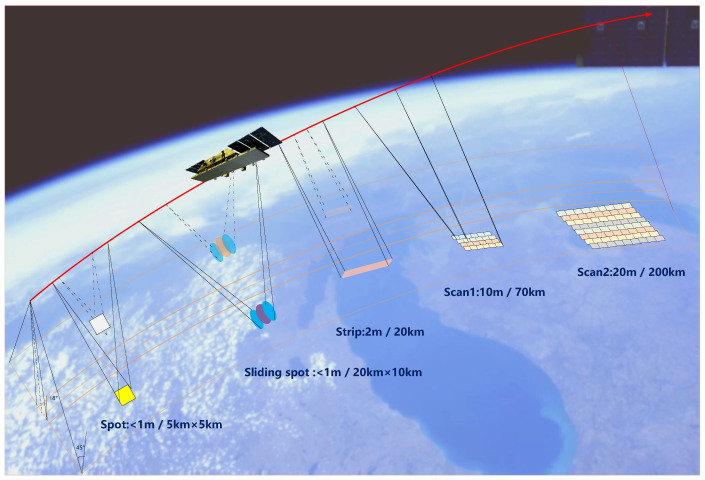
SAR working modes.

**Figure 5 sensors-24-06741-f005:**
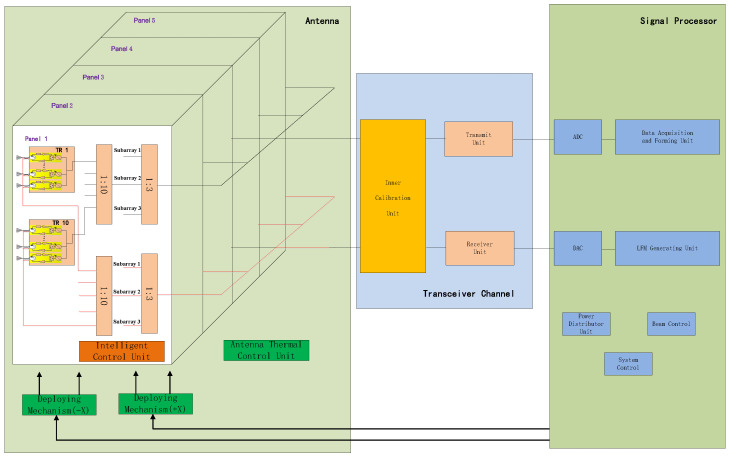
The electric block diagram of the SAR payload system.

**Figure 6 sensors-24-06741-f006:**
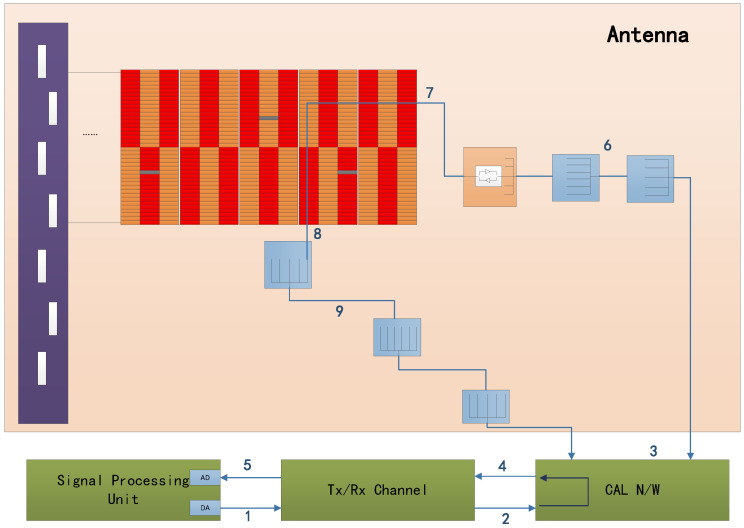
The calculated network of antenna.

**Figure 7 sensors-24-06741-f007:**
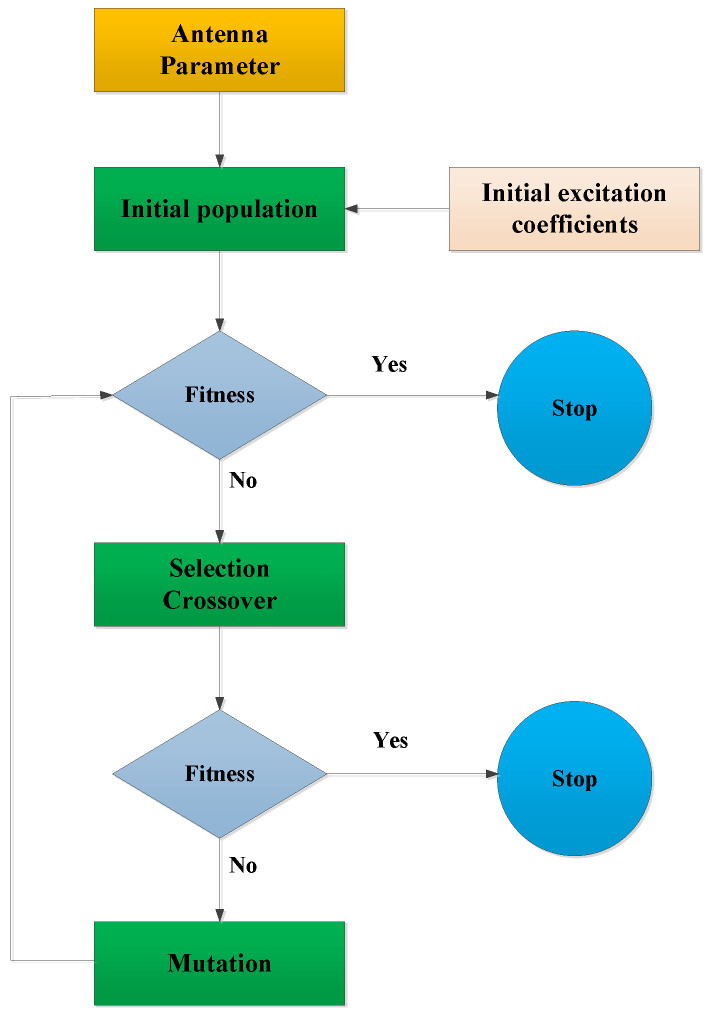
Beam optimization technique flow chart.

**Figure 8 sensors-24-06741-f008:**
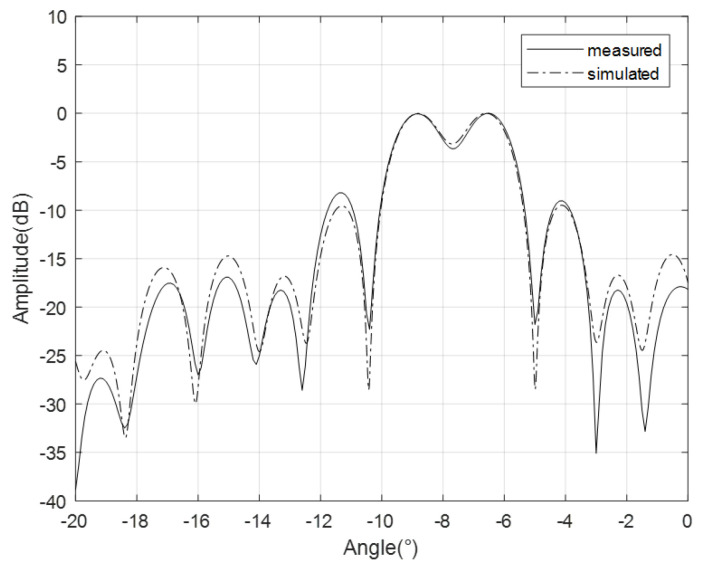
Ku−band antenna pattern validation.

**Figure 9 sensors-24-06741-f009:**
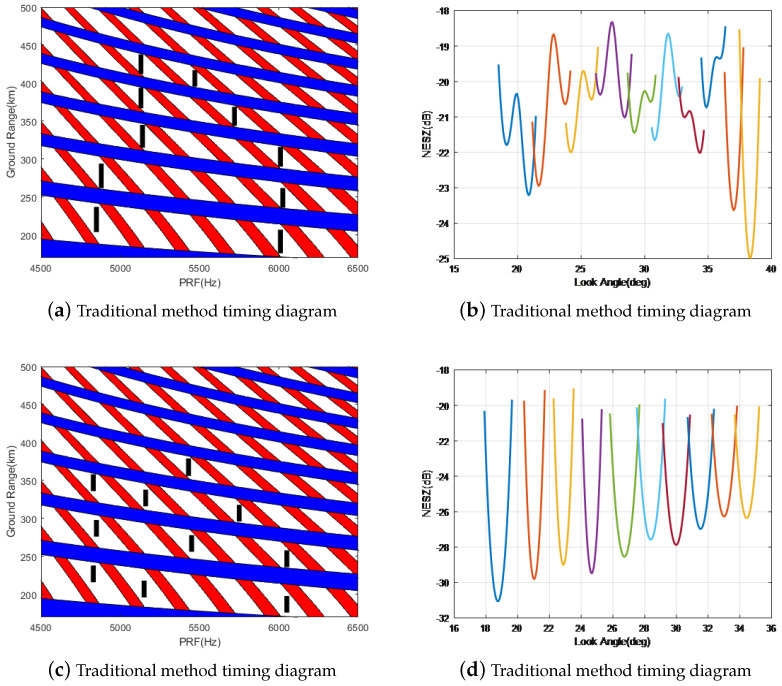
Ku−band SAR system performance.

**Figure 10 sensors-24-06741-f010:**
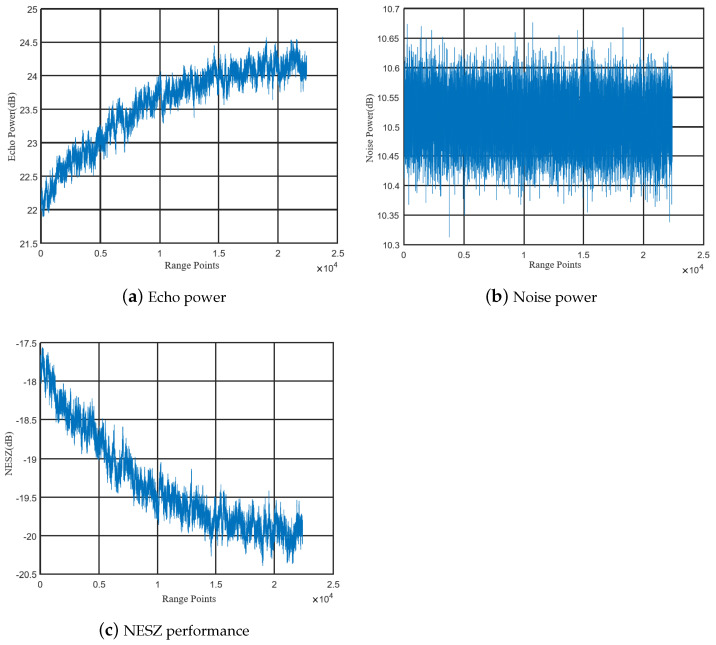
NESZ as assessed from the Amazon rainforest.

**Figure 11 sensors-24-06741-f011:**
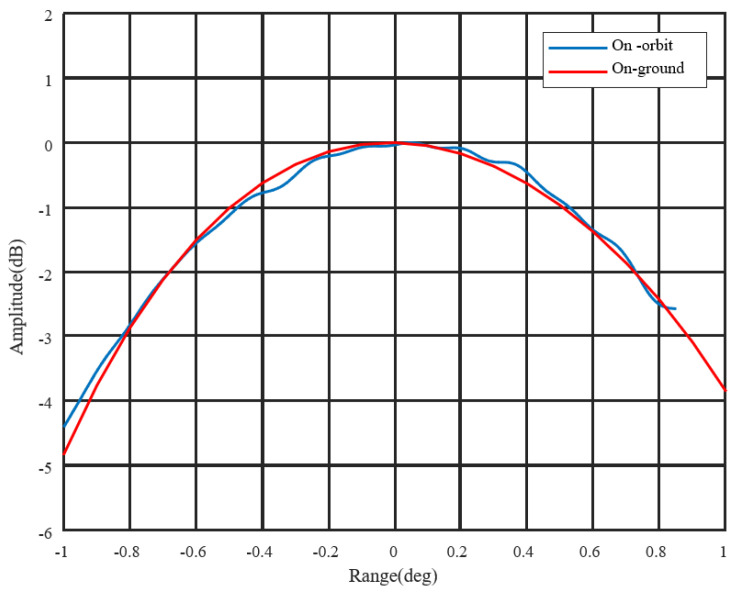
Antenna pattern as assessed from the Amazon rainforest.

**Figure 12 sensors-24-06741-f012:**
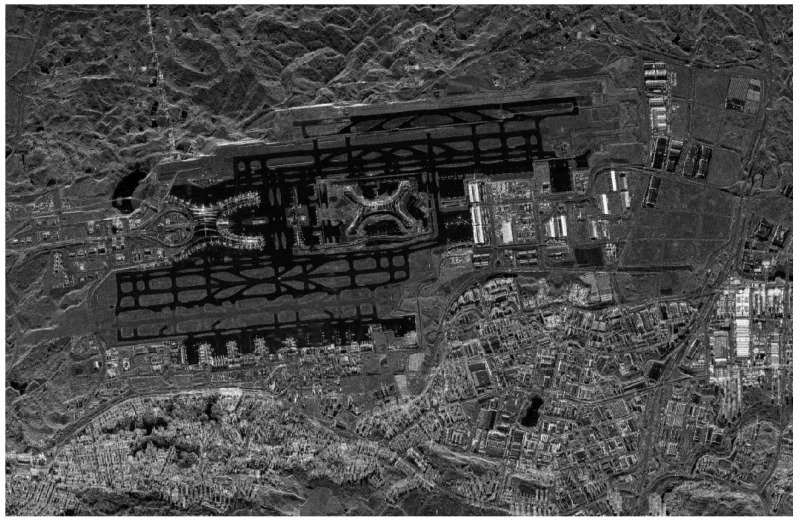
Chongqing Airport, China (strip mode, photographed at 3:30 on 27 January 2024 (UTCG)).

**Figure 13 sensors-24-06741-f013:**
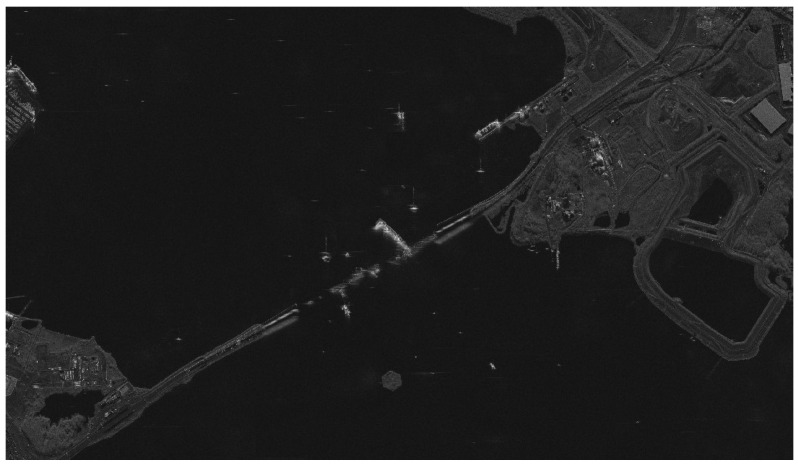
Francis Scott Key Bridge, USA (sliding spot mode, photographed at 16:19 on 30 March 2024 (UTCG)).

**Figure 14 sensors-24-06741-f014:**
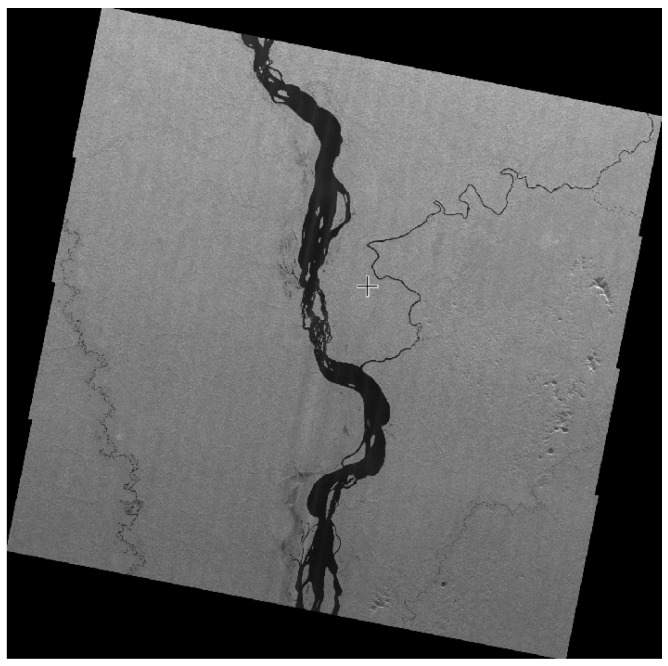
Brazilian Amazon Rainforest (Scan 1 mode).

**Figure 15 sensors-24-06741-f015:**
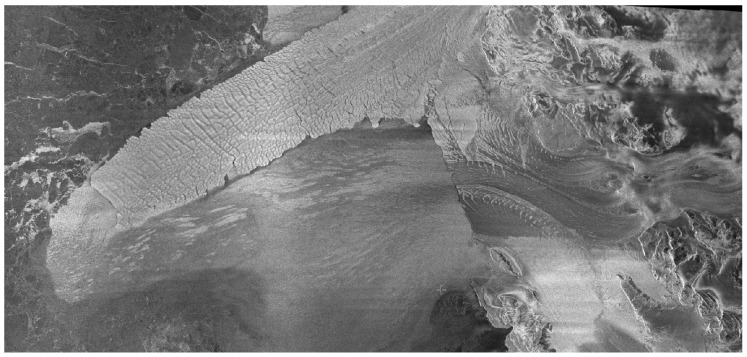
Antarctic glacial tongue (Scan 2 mode).

**Table 1 sensors-24-06741-t001:** Ku-band SAR payload key parameters.

Parameter	Value
Orbit Height	530 km
Orbit Inclination	97.5°
Descending node time	10:30 AM
Band	Ku
Polarization	VV
Antenna Size	3.6 m × 0.6 m
Number of Panels	5
Antenna Columns (Azimuth)	15
Antenna Rows (Elevation)	40
Peak Power	4000 W
Average Power Consumption	3200 W max
Pulse Width	10–40 us
Signal Bandwidth	20–600 MHz
Transmit Duty Cycle	20% max
Receiver Noise Figure	3.2 dB max
Pulse Repetition Frequency	4000–8000 Hz (programmable)
NESZ (Noise Equivalent Sigma Zero)	<−18 dB
Output Data Rate	4.8 Gbps max
Payload power consumption	<3200 W
Payload Mass	79.8 kg

**Table 2 sensors-24-06741-t002:** The comparison of SAR key parameters.

	ICEYE	TerraSAR	Taijing 4(03)
Frequency band	X	X	Ku
Slidspot mode	1 m/5 × 5 km	1 m/10 × 10 km	1 m/20 × 10 km
Strip mode	3 m/30 km	3 m/30 km	2 m/20 km
Scan mode	15 m/100 km	Scan 1: 16 m/100 km	Scan 1: 10 m/70 km
		Scan 2: 40 m/200 km	Scan 2: 20 m/200 km
NESZ	−15 dB	−19 dB	−18 dB
AASR	−20 dB	−20 dB	−20 dB
RASR	−20 dB	−20 dB	−20 dB
Mass	<100 kg	1230 kg	230 kg
		(394 kg for payload)	(80 kg for payload)

## Data Availability

Data are contained within the article.

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
