# Peer review of "Design and On-Orbit Performance of Ku-Band Phased-Array Synthetic-Aperture Radar Payload System"

_sensors, 2024, doi:10.3390/s24206741_

Round 1
Reviewer 1 Report
Comments and Suggestions for Authors
This paper introduces the development and results of Ku-band phase-array SAR system. I have the following questions.
1. The paper is written in a technique report style. Thus it is difficult to find the most important work in the development of the system. It seems the beampattern optimization is of critical important. However, the proposed optimization method contains very limited novelty. So, it is suggested to provide more insightful statement on the key techniques to support this paper. For example, is there range ambiguity problem and how to suppress the range ambiguity? By using waveform diversity or by using DBF?
2. The results should be improved. For example, the normalized equivalent scattering is not provided, thus the result is misleading. The x-axis is indicated as Range(deg), what is the meaning? Y-axis is indicated as dB, what is the physical quantity?
Comments on the Quality of English Languagesatified
Reviewer 2 Report
Comments and Suggestions for Authors
The manuscript presents an interesting study on the design and on-orbit performance of a novel Ku-band phased-array SAR payload. However, the current version requires significant improvements in technical detail, comparative analysis, and clarity of presentation. Incorporating the suggested references and addressing the identified issues will enhance the overall quality and impact of the paper.
1:The manuscript lacks a comprehensive comparison of the proposed Ku-band SAR system with existing technologies in terms of performance metrics like range ambiguity, azimuth ambiguity, and system sensitivity. To provide a clearer context for your work's novelty and value, include a comparison with other SAR systems, particularly those utilizing similar phased-array technology. For example, refer to “An advanced scheme for range ambiguity suppression of spaceborne SAR based on blind source separation” to discuss how your method compares in terms of range ambiguity suppression.
2:The description of the beam optimization technique is vague and lacks mathematical rigor. The authors should provide more detailed equations and explanations to clearly illustrate how the optimization was achieved, particularly regarding the criteria for selecting the initial excitation coefficients and the method used for fitness function calculation. Including more in-depth analysis or simulation results would strengthen the manuscript. Referencing “Sub retrograde geosynchronous orbit SAR: parameter design and performance analysis” could provide useful insights into presenting detailed parameter optimization techniques.
3: The manuscript describes on-orbit performance but does not sufficiently detail the testing methodology. The process of verifying system sensitivity, beam patterns, and image quality using the Amazon rainforest should be explained more thoroughly. For example, specify the exact metrics used to evaluate the SAR images and the statistical significance of the results obtained. Additionally, provide more information on the environmental conditions during these tests, such as orbital parameters and atmospheric interference, to better support the claimed performance improvements.
4:The payload's key parameters are presented in Table 1, but several values need clarification. For instance, the rationale behind choosing specific pulse widths, bandwidths, and the number of panels in the phased array should be explicitly stated. Explain why a peak transmission power exceeding 4000W was necessary and how this impacts the satellite's overall power budget. Furthermore, provide more context on how these parameters influence the payload’s operational modes and performance.
5:While the manuscript outlines various operating modes (e.g., spot, sliding spot, strip, scan 1, scan 2), it does not adequately discuss the trade-offs between these modes, such as resolution versus coverage, and how these trade-offs are managed in practical scenarios. A more detailed analysis should be included to highlight how the choice of mode affects the overall mission objectives and data quality.
6: Some figures, such as Figures 4, 6, and 8, lack detailed captions and explanations, which makes it difficult to understand the context and relevance of the data presented. Enhance the figure descriptions to clarify the content, such as specifying what each axis represents and what conclusions can be drawn from the data.
7:The mathematical models presented in Section 3.1 are not clearly defined. Terms such as "θEl" and "θAz" are introduced without sufficient explanation. Provide a clearer definition of all variables and parameters in the equations, and include a brief overview of the physical meaning and importance of each term.
Reviewer 3 Report
Comments and Suggestions for Authors
The authors presented a paper titled "Design and on-orbit performance of Ku-band phase-array SAR payload system" my suggestions are as follows:
1. Kindly provide a comparison table by comparing important parameters with existing literature
2. In Figure 6, what does the numbers 1, 2, 3... mentioned in the figure indicate. kindly explain this figure in more detail
3. Kindly provide detailed explanation of Figure 7.
Round 2
Reviewer 2 Report
Comments and Suggestions for Authors
accept
Reviewer 3 Report
Comments and Suggestions for Authors
Authors have answered all the queries, hence my decision is to accept the paper